# In situ and remote observations of the ultraviolet footprint of the moon Callisto by the Juno spacecraft

J. Rabia [1] ✉, V. Hue [2], C. K. Louis [3], N. André[1,4], Q. Nénon[5], J. R. Szalay [6], R. Prangé [3], L. Lamy [3,7], P. Zarka [3], B. Collet [7], F. Allegrini[8,9], R. W. Ebert [8,9], T. K. Greathouse [8], B. Bonfond [10], G. R. Gladstone [8], A. H. Sulaiman [11], W. S. Kurth [12], J. E. P. Connerney[13,14], P. Louarn[1], E. Penou[1], A. Kamran [1], D. Santos-Costa[8], R. S. Giles[8], J. A. Kammer[8], M. H. Versteeg[8] & S. J. Bolton [8]

Jupiter exhibits peculiar multiwavelength auroral emissions resulting from the electromagnetic interactions of Io, Europa, and Ganymede with the magnetospheric plasma flow. Characterizing the faint auroral footprint of the fourth Galilean moon, Callisto, has always been challenging because of its expected weakness and its proximity to Jupiter's bright main aurora. Here, we report on unusual magnetospheric conditions that led to an equatorward shift of Jupiter's main auroral oval unveiling the auroral footprints of the four Galilean moons in a single observation. Remote observations by the Juno spacecraft reveal a double-spot structure, characteristic of the footprints of the other three moons, with a maximum ultraviolet brightness of $137 \pm 15$ kR. Concurrent observations within Callisto's flux tube reveal field-aligned electrons with a characteristic energy of 10 keV, depositing an energy flux of 55 mW.m$^{-2}$ in Jupiter's atmosphere. The electron properties are consistent with the triggering of radio emissions with intensities lower than $5 \times 10^{-18}$ W.m$^{-2}$.Hz$^{-1}$.

Like Earth, Jupiter's atmosphere hosts a variety of auroral emissions observed in ultraviolet (UV), infrared (IR), X-ray, visible, and radio. At both planets, these emissions have time-variable structures and dynamics, with the brightest structures produced by accelerated electron fluxes associated with magnetic-field-aligned currents. The main power source for auroral emissions is the solar wind interaction at Earth and the planetary rotation at Jupiter. The brightest structure at Jupiter takes the form of a main auroral oval that is related to a magnetosphere-ionosphere current system coupling the magnetodisk plasma with Jupiter's ionosphere, and associated with the departure from corotation of the plasma at radial distances of about 20–40 $R_J$[1] (1 $R_J = 71,492$ km = Jupiter's equatorial radius). Peculiar features of Jupiter's auroral emissions, with no counterpart at Earth, are the auroral structures induced by the interaction of the Galilean moons Io, Europa, and Ganymede with Jupiter's magnetosphere. These footprints are observed in the UV[2–4], IR[5–7], and radio[8–12] wavelengths as the result of

[1]Institut de Recherche en Astrophysique et Planétologie (IRAP), CNRS, CNES, Toulouse, France. [2]Aix-Marseille Université, CNRS, CNES, Institut Origines, LAM, Marseille, France. [3]LIRA, Observatoire de Paris, Université PSL, Sorbonne Université, Université Paris Cité, CY Cergy Paris Université, CNRS, 92190 Meudon, France. [4]ISAE-Supaero, Université de Toulouse, Toulouse, France. [5]Laboratoire Atmosphère Observations Spatiales, CNRS, Sorbonne Université–CNES, Paris, France. [6]Department of Astrophysical Sciences, Princeton University, Princeton, NJ, USA. [7]Aix-Marseille Université, CNRS, CNES, LAM, Marseille, France. [8]Southwest Research Institute, San Antonio, TX, USA. [9]Department of Physics and Astronomy, University of Texas at San Antonio, San Antonio, TX, USA. [10]Laboratory for Planetary and Atmospheric Physics, University of Liège, Liège, Belgium. [11]School of Physics and Astronomy, Minnesota Institute for Astrophysics, University of Minnesota, Minneapolis, MN, USA. [12]Department of Physics and Astronomy, University of Iowa, Iowa City, IA, USA. [13]Space Research Corporation, Annapolis, MD, USA. [14]NASA/Goddard Space Flight Center, Greenbelt, MD, USA. ✉e-mail: jonas.rabia@irap.omp.eu

the local sub-Alfvénic electromagnetic interaction, with Alfvén Mach numbers ranging from 0.3 to 0.7[13], between each moon and the magnetospheric plasma flow. While UV and IR auroral emissions are induced by electron precipitations within Jupiter's atmosphere, moon-induced radio emissions result from unstable electron populations propagating away from Jupiter. Electrons responsible for these multi-wavelength emissions are accelerated by Alfvén waves and/or electric currents generated by the local interaction between the magnetospheric plasma flow and the moons. As the corotation breakdown is expected to occur at radial distances larger than the orbital distances of Io, Europa, and Ganymede, their UV and IR auroral footprints are located equatorward from the main auroral oval on Jupiter's polar regions.

Since July 2016, these interactions and the resulting auroral emissions have been characterized in detail by the Juno mission[14] thanks to its highly-eccentric and polar orbit around Jupiter. The spacecraft enables both remote-sensing observations, including UV, IR, and radio measurements, and in situ observations, with plasma and wave measurements obtained along the magnetic flux tubes connecting the moons to their auroral footprints. These observations confirmed that electrons and protons are accelerated over a broad energy range[15–22], mainly by inertial Alfvén waves for electrons[23–25] and resonant interactions with ion-cyclotron waves for protons[19,23].

Among the four Galilean moons, the case of Callisto remains poorly documented despite a remote detection of its footprint[26] based on Hubble Space Telescope ultraviolet observations. However, the lack of multiple detections does not allow a complete characterization of its properties. More context on these previous detections is provided in Supplementary Fig. 1, where we used improvement of magnetic field models enabled by Juno to revisit the mapping of the previously detected spots. In IR and radio, no clear observation of Callisto-induced auroral emissions has been reported. These detections are particularly challenging due to the location of Callisto's weak footprint buried within the bright main auroral oval[27]. Callisto's interaction with Jupiter's magnetosphere also differs from that of the other three Galilean moons, as the moon encounters super-Alfvénic local electromagnetic interactions when located at the center of the plasma sheet[13,28]. As this interaction is only slightly super-Alfvénic, no clear bow shock structure is expected to form[29]. Still, even if a bow shock does not form, the energy transfer efficiency between the Callisto environment and Jupiter's atmosphere may be significantly reduced[30]. However, due to the tilt between the centrifugal equator and the moon's plane of rotation, Callisto is subject to a highly-variable plasma and magnetic environment during its orbit around Jupiter[31]. This enables sub-Alfvénic interactions to develop when the moon is located below or above the plasma sheet, where the Alfvén speeds are much higher than the plasma flow[28].

Here, we show that unusual magnetospheric conditions led to a significant shift of the location of the main auroral oval in the upper atmosphere of Jupiter. Such shifts have been previously observed[32–34], and were also suggested in relation to the previous detection of the Callisto footprint[26]. This allowed the auroral footprints of the four Galilean moons to be revealed in a single observation by Juno, enabling the precise characterization in UV, radio, plasma, and waves of the high-latitude signatures of the Callisto-magnetosphere interactions. Comprehensive cross-comparisons of the plasma interactions between the Galilean moons and Jupiter's magnetosphere can be performed.

## Results

### Expansion of Jupiter's magnetosphere
During its 22nd perijove (PJ22), the Magnetic Field investigation (MAG[35]) onboard Juno measured a significant change in the state of the middle magnetosphere of Jupiter with a current intensity $\mu_0 I_{MD}/2$ for the magnetized disk of plasma surrounding Jupiter estimated at 156.1 nT,

where $\mu_0$ and $I_{MD}$ are the vacuum permeability and the current intensity in the magnetodisk, respectively. This value is the highest reported for Juno's first 24 orbits, compared to the mean value of 140.5 nT, and well above the $1\sigma$ standard deviation of 8.25 nT[36]. In addition, the Ultraviolet Spectrograph (UVS[37]) onboard Juno simultaneously observed the position of the auroral main oval during PJ22, and measured an equatorward expansion of $1800 \pm 300$ km with respect to the reference oval[32].

Based on a solar wind propagation model[38], we inferred that the solar wind dynamic pressure during Juno's PJ22 was $1–3 \times 10^{-2}$ nPa and remained steady for a few days around the perijove (Supplementary Fig. 2). We note that during this time, the longitude separation between the Earth and Jupiter is ~80°, providing good confidence in the solar wind propagation model. With this weak dynamic pressure range, below the 10th percentile occurrence probability[39,40], we estimate that the standoff distances to the Jovian magnetopause and bow shock are 95–110 $R_J$ and 124–143 $R_J$, respectively, i.e., larger than their average values of 75 $R_J$ and 84 $R_J$[39]. The measured expansion of the auroral main oval, the increase in the magnetospheric current constant, combined with the weak solar wind dynamic pressure during PJ22 all indicate that the magnetosphere was expanded during Juno's PJ22. The strong correlation between the expansion of the main oval and the magnetospheric expansion[32] may be explained by an outward stretching of the magnetic field lines, resulting in an equatorward shift. This effect has been studied using both theoretical and numerical models[41–43], as well as ultraviolet observations[32–34].

Partial maps of the auroral structures observed by Juno-UVS on Jupiter's northern pole during PJ22 (2019-09-12) are displayed on Fig. 1b. Juno-UVS revealed a main auroral oval significantly shifted equatorward with respect to the reference location, close to the reference auroral footpath of Ganymede, mapping to about 15 $R_J$, and, in some locations, skimming close to the footpath of Europa, mapping to about 9.4 $R_J$ in the equatorial plane. The footpaths of the moons correspond to a sequence of magnetic projections of the moons' location along their orbits onto Jupiter's atmosphere, forming an oval-shaped contour. The moons' footpaths therefore indicate the statistical location of the moons' auroral footprints. The significant shift of the main oval is consistent with the expansion of the Jovian magnetosphere due to varying solar wind pressure[44], as previously described. This expansion is also coherent with an increase of current intensity, as observed by Juno-MAG.

### UV footprints of the four Galilean moons
As Juno skimmed over Jupiter's poles during PJ22, the Juno-UVS instrument scanned the auroral regions of Jupiter using the 30-s spacecraft spin. The UV auroral footprints of Io, Europa, and Ganymede are identified[45] very close to their expected reference auroral footpaths, as observed in Fig. 1b. This suggests that the topology of the magnetic field lines connecting the moons to Jupiter remains the same under the magnetospheric expansion, and that the expansion of the main emission results from a radial shift of the source of the emission, at radial distances below 26 $R_J$. The associated UV emissions for each moon consisted of two bright spots associated with a Main Alfvén Wing (MAW) spot and a Transhemispheric Electron Beam (TEB) spot[18,46], respectively. In the case of a moon located at the center of the plasma sheet, the TEB and MAW spots are co-localized[47], as observed here for Europa. For Io, we expect the TEB and MAW spots to be separated by a few degrees, but to appear co-located because of the time integration and spatial resolution of this observation. This pair of spots is usually followed by a faint auroral tail[48]. Here, we identify the auroral tails of Io and Europa, as well as possibly for Ganymede, although buried within the main auroral oval.

The predicted footpath of Callisto, displayed as a green dashed line in Figs. 1b–2a, is no longer co-located with the main auroral oval (Supplementary Fig. 4). A pair of spots is clearly observed along this

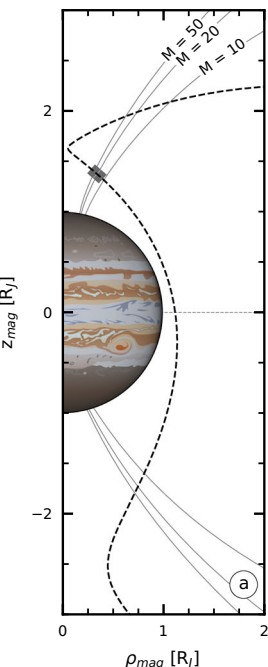

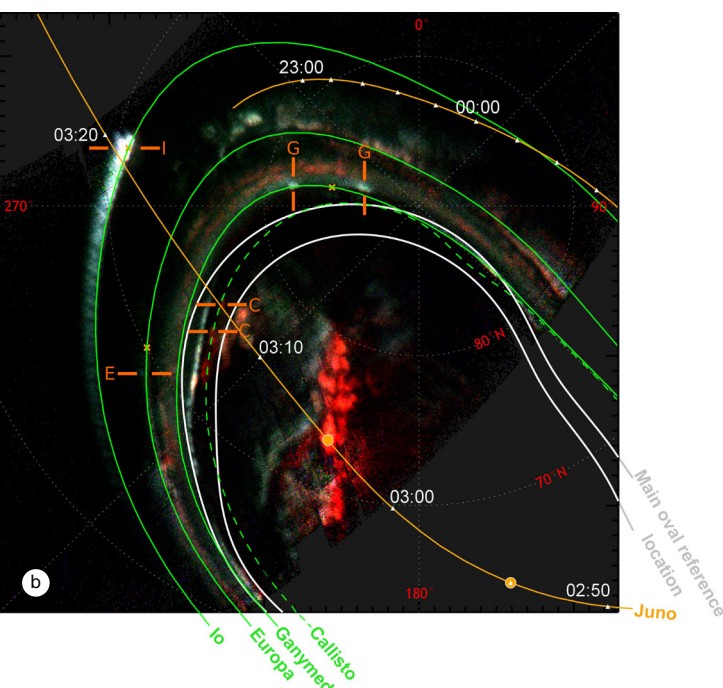

**Fig. 1 | Juno-UVS observations of the northern auroral region of Jupiter during PJ22. a** Juno trajectory plotted in magnetic coordinates. The gray area indicates where the in situ measurements shown in Fig. 3 have been made. **b** False color UV map of the auroral structures observed onto Jupiter's northern auroral region resulting from co-adding consecutive Juno-UVS data from 02:54:00 to 03:09:02. The colors represent various UV spectral bands: red, green, and blue tend to correspond to high-, medium-, and low-energy electron precipitation, respectively, while white indicates a mixture of energies[64,86]. The magnetic footpaths of the Juno spacecraft and the Galilean moons are shown as orange and green lines, respectively. The auroral footpaths of Io, Europa, and Ganymede (green solid lines) were calculated using JRM33 + CON2020[36,87] while the Callisto footpath (green dashed line) was derived using JRM33 + KK2005[75] (see "UV maps" in Methods). The white triangles along the Juno footpath highlight Juno's magnetic footprints with a 10 min time step. Juno's magnetic footprints at the beginning and end of the Juno-UVS data integration time are represented by orange dots. The white boundaries show the statistical position of the main oval emissions. Orange crosses indicate the statistical location of the Main Alfvén Wing (MAW) spots of Io, Europa, and Ganymede[45]. The footprints of the four Galilean moons are outlined by orange lines. The un-annotated Juno-UVS observation is displayed in Supplementary Fig. 3.

footpath and interpreted as footprints of Callisto-induced UV emissions, that we refer to as leading (uppermost spot along the Callisto footpath in Figs. 1b–2a) and trailing spots (lowermost spot along the Callisto footpath in Figs. 1b–2a). In this observation sequence, Juno-UVS scanned across the auroral region near the expected Callisto auroral footprint, resulting in a partial coverage of the two spots over the considered sequence. The calculation of the rotation rate of the leading and trailing spot leads to $1.01 \pm 0.49 \left[ \times 10^{-2} deg.s^{-1} \right]$ and $0.98 \pm 0.29 [\times 10^{-2} deg.s^{-1}]$, respectively. This confirms that they do not co-rotate with the magnetosphere of Jupiter ($d\lambda_{III}/dt = 0$ deg.s$^{-1}$) but rather closely follow the orbital motion of Callisto, whose rotation rate is given by $d\lambda_{Cal}/dt = 360°/P_{syn}^{Cal} = 0.98 [\times 10^{-2} deg.s^{-1}]$ (Fig. 2b, Methods). Based on these calculations, we confirm that the spots identified are auroral emissions induced by Callisto. A similar calculation based on observations of the Cassini Ultraviolet Imaging Spectrograph (UVIS)[49] led to the discovery of Enceladus' UV auroral footprints in the upper atmosphere of Saturn[50,51].

At 03:13 UT, when Juno is connected to the leading spot of the Callisto footprint, the moon is located below the equatorial magnetized disk of plasma, with magnetic and centrifugal latitudes equal to -3.8° and -2.8°, respectively. The orbital parameters of the Galilean moons and Juno at that time are provided in Supplementary Table 1. In such an orbital configuration, where Callisto is below the plasma sheet and sub-Alfvénic Mach number conditions prevail, an Alfvén wing structure forms and leads to a pair of spots, as previously observed for the other three Galilean moons. It is expected that the TEB spot, created by electrons accelerated in Jupiter's southern hemisphere away

from the planet and precipitating into the northern hemisphere, is located upstream of the MAW spot[18,52] (Supplementary Fig. 5). The leading and trailing spots observed in the Callisto footprints are therefore associated with a TEB spot followed by a MAW spot, respectively. We note that the longitude separation existing between the TEB and MAW spots also indicates that a plasma dense enough to slow down the propagation of Alfvén waves exists in Callisto's wake (see "Estimate of the electron density at Callisto" in Methods), as previously observed by the Galileo spacecraft[53,54]. By studying the longitude separation between the TEB and the MAW spots and their theoretical instantaneous positions, we have estimated the total plasma and electron density and the plasma scale height at Callisto's orbital location (see "Estimate of the electron density at Callisto" in Methods). The electron density derived at Callisto, i.e., $n_{at Callisto}^{electrons} = 0.1 \pm 0.01$ cm$^{-3}$, is within the range of those measured at Callisto by the Galileo spacecraft[31], i.e., $n_e = 0.01 - 0.7$ cm$^{-3}$. We estimate that the plasma density at the center of the plasma sheet at Callisto's orbital distance was $n_0^{electrons} = 0.15 \pm 0.02$ cm$^{-3}$ and the scale height $H = 0.94$ $R_J$ at the time of the Callisto footprint observation, in agreement with Juno-derived scale height measurements (0.7–2.4 $R_J$)[55], though lower than derived from Voyager and Galileo data (3.5 $R_J$)[56].

The UV brightness associated with the TEB and MAW spots are $108 \pm 11$ kR and $137 \pm 15$ kR, respectively. These values are lower than those observed for the footprints of Io (2000 kR)[57], Europa (180 kR)[48], and Ganymede (900 kR)[48], confirming the weak brightness of the Callisto footprint. In the UV false color images displayed in Fig. 1b, the Callisto footprint appears redder compared to the footprints of Io,

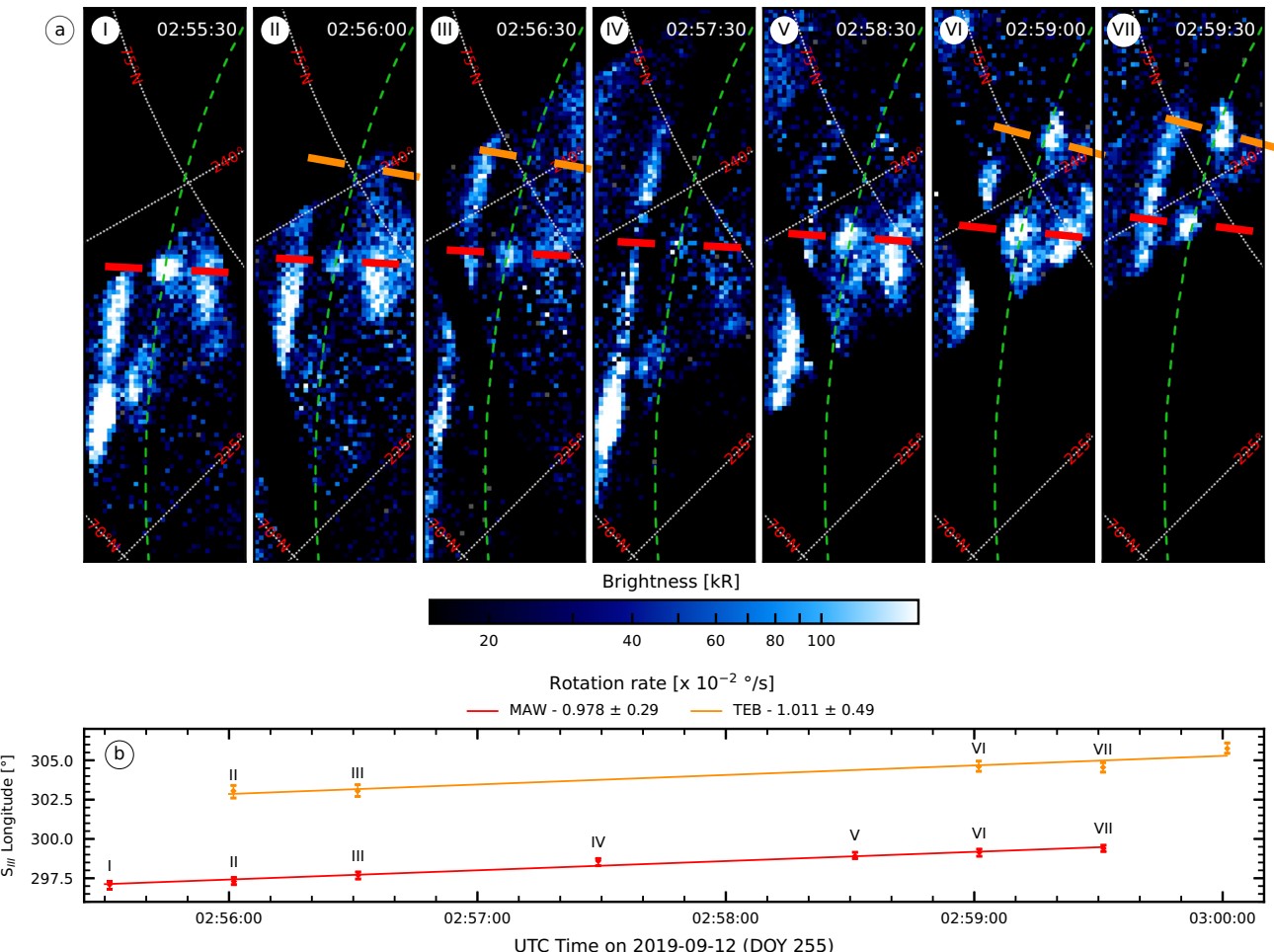

**Fig. 2 | Time evolution of the Callisto footprint. a** Juno-UVS spin-by-spin observations of the Callisto UV footprint. The leading and trailing spots, identified as Transhemispheric electron beam (TEB) and Main Alfvén Wing (MAW) spots, are highlighted by orange and red lines, respectively, when observed. The dashed green line represents Callisto's footpath derived using the JRM33 + KK2005 magnetic field model. Each frame is centered around the same $S_{III}$ longitude/ latitude, shown as a white dotted grid. In this reference frame, the Callisto footprint gradually drifts over time. Conversely, nearby auroral emissions, co-rotating with Jupiter's magnetosphere, are fixed over time. **b** Evolution of the System-III ($S_{III}$) equatorial longitude of the TEB (orange) and MAW (red) spots of Callisto as a function of time.

Europa, and Ganymede, with false color intensities similar to those of the polar emissions. In the similar way of the color ratio, this parameter reflects the contribution of the methane absorption in the auroral emission, thus usually providing information on the depth of the auroral emission (see "UV maps" in Methods). However, in the main oval region, close to the location of the Callisto footprint, the vertical altitude distribution of the methane is significantly different from that encountered in the regions where the footprints of Io, Europa, and Ganymede are located[58,59]. As the methane homopause altitude was observed to be higher in the polar auroral region[58,59], we suggest that auroral emissions triggered at the same altitude will appear redder (or with a higher color ratio) in this region, as observed for the Callisto footprint. However, we can not rule out that the characteristic energy of the electrons inducing Callisto's footprint is higher than for Io and Europa, explaining the higher color ratio, as sometimes observed on Ganymede's footprints.

**In situ electron and waves observations**

During PJ22, the Jovian Auroral Distributions Experiment (JADE)[60] instrument measured electrons in the 50 eV–72 keV energy range above Jupiter polar caps. Consisting of two electron sensors which provide a 240°-field of view, a partial or complete coverage of the pitch angle can be obtained every second when the instrument is operating in high-resolution science mode. At the same time, the Juno-Waves instrument[61] carried out a survey of the radio and plasma waves using a single axis electric antenna and a single axis magnetic field sensor. Juno-Waves can simultaneously measure both the electric and magnetic fields in the 50 Hz–20 kHz frequency range[61] at a 1-s resolution. Electric field measurements can be performed up to 40 MHz.

A part of the electron observations as well as waves measurements made during PJ22N is presented in Fig. 3. This corresponds to a time interval when Juno crossed the Callisto footpath (Figs. 1b–2a).

Juno-JADE-E observations show two consecutive increases of the electron flux (Fig. 3a), corresponding to precipitating electrons (Fig. 3b). During this time interval, Juno was first magnetically connected to the polar cap, then to the M-shell of Callisto, and finally to the main auroral oval (Fig. 1b). Along magnetic field lines connected to the Callisto footprint, field-aligned electrons with energy below 30 keV are measured, filling the downward loss cone. The resulting downward electron energy flux and characteristic energy are estimated to be 55 mW.m$^{-2}$ and 10 keV, respectively. This is consistent with the electron properties reported during crossings of field lines connected to the TEB spot of Ganymede[18,21] and Europa[17,20], with larger downward energy fluxes and characteristic energy than those reported in the MAW and auroral tail flux tubes[15,20,21]. The in situ properties of the electrons inducing Io's TEB spot have not yet been reported.

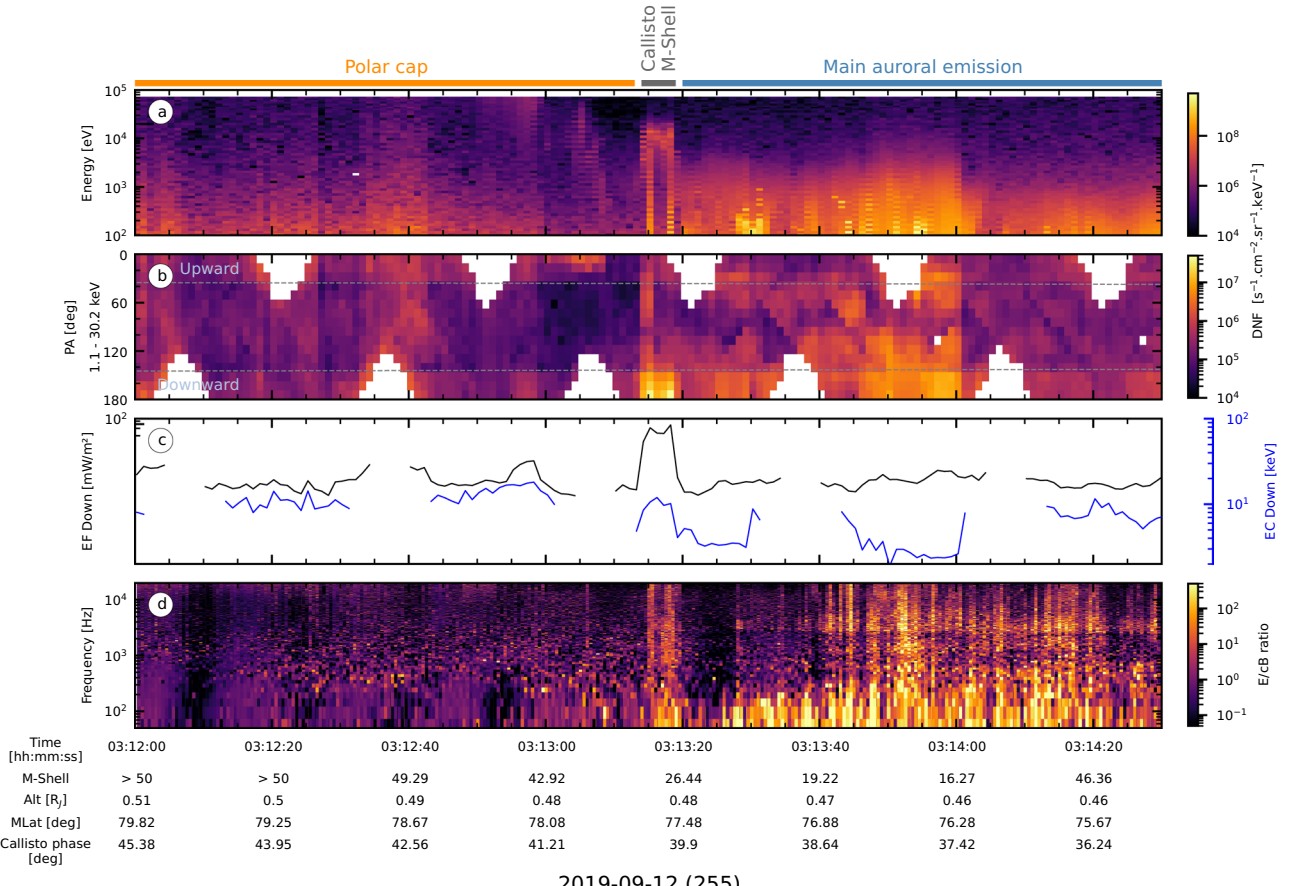

**Fig. 3 | JADE-E and Waves in situ observations. a** Electron energy-time spectrogram. **b** Pitch angle (PA)-time spectrogram for electrons with energies between 1.1 and 30.2 keV. The size of the loss cone is indicated by the dashed gray lines. **c** Partial electron downward energy flux (EF Down, in black) and characteristic energy (EC Down, in blue). Electrons with energy between 1.1 and 30.2 keV have been considered. **d** Frequency-time spectrogram of the ratio of the electric (E) to magnetic (cB) spectral densities. Color bands above (**a**) indicate on which auroral structure on Jupiter's northern pole Juno's magnetic footprint maps. Juno's M-Shell, altitude (Alt), magnetic latitude (MLat), and longitude separation to Callisto as a function of time are indicated below the panel (**d**).

Simultaneously with the electron beam measurement, a sharp increase in electric and magnetic spectral densities is observed over the full frequency range sampled (Supplementary Fig. 6). High value ratios of electric and magnetic spectral densities (Fig. 3d) during the flux tube crossing, i.e. $\delta E >> c\delta B$, suggest quasi-electrostatic waves rather than electromagnetic waves, as also previously observed in the flux tubes of Ganymede[18] and Enceladus[62,63]. These structures, known as large-amplitude electrostatic solitary waves (ESWs), are highly correlated with energetic electron fluxes[64].

Figure 4a compares the downward electron energy distribution measured within the Callisto flux tube (colored lines) and outside of it (gray lines). The former corresponds to a peaked distribution, with a differential number flux (DNF) increase confined in a narrow energy range, as shown in Fig. 3a and as observed within Europa and Ganymede's TEB flux tubes[17,18,20,21]. This distribution is far from the broadband electron energy distributions, meaning that the increase in electron DNF is observed over a wide energy range, as previously observed in the flux tubes connected to the MAW spots and auroral tails of the Io, Europa, and Ganymede footprints[15,20,21]. Juno-JADE-E in situ observations are therefore consistent with a crossing of the magnetic field line connected to the TEB spot of Callisto, remotely observed by Juno-UVS.

**Implications for Callisto-induced radio emissions**

Radio-emissions associated with the moons Io[8], Europa[9,12], and Ganymede[10,65] have been observed from both space- and ground-based observatories. These emissions are produced at high latitude above Jupiter's atmosphere along the Galilean moons' flux tube by the Cyclotron Maser Instability[66] (CMI). Generated by weakly relativistic electrons, these radio emissions are produced at a frequency close to the local electron cyclotron frequency. Analysis based on Juno-Waves data revealed that the sources of the moon-induced radio emissions can be located on magnetic field lines connected to the TEB spot, MAW spot, and auroral tail of the Io, Europa, and Ganymede footprints (ref. 11 and references within). Hints of Callisto-induced radio emissions were found in Galileo and Voyager data[67,68] but have been later attributed to biases in data selection and processing[12]. Thus, no unambiguous observations have been reported so far. In Fig. 4b, we show the electron distribution function (EDF) in the velocity space derived from the Juno-JADE-E data within the Callisto flux tube. A CMI growth rate analysis applied to this EDF reveals radio waves can be amplified along a resonance circle tangent to the loss cone. Indeed, it can be seen that inside the upward loss cone, the iso-energy contours (black lines) do not follow the shape of a maxwellian, i.e., a half circle in the EDF, indicating the presence of unstable loss-cone electron populations. This is similar to the source EDF for decametric emission driven by Io, Europa, and Ganymede, i.e., an unstable loss-cone population of upgoing electrons[9–11].

The calculation of the growth rates associated with these electron distribution functions (see "Cyclotron Maser Instability and growth rate calculation" in Methods) gives values high enough (>10^{-6}) to amplify radio emissions in the Callisto flux tubes at 7.75 MHz over the

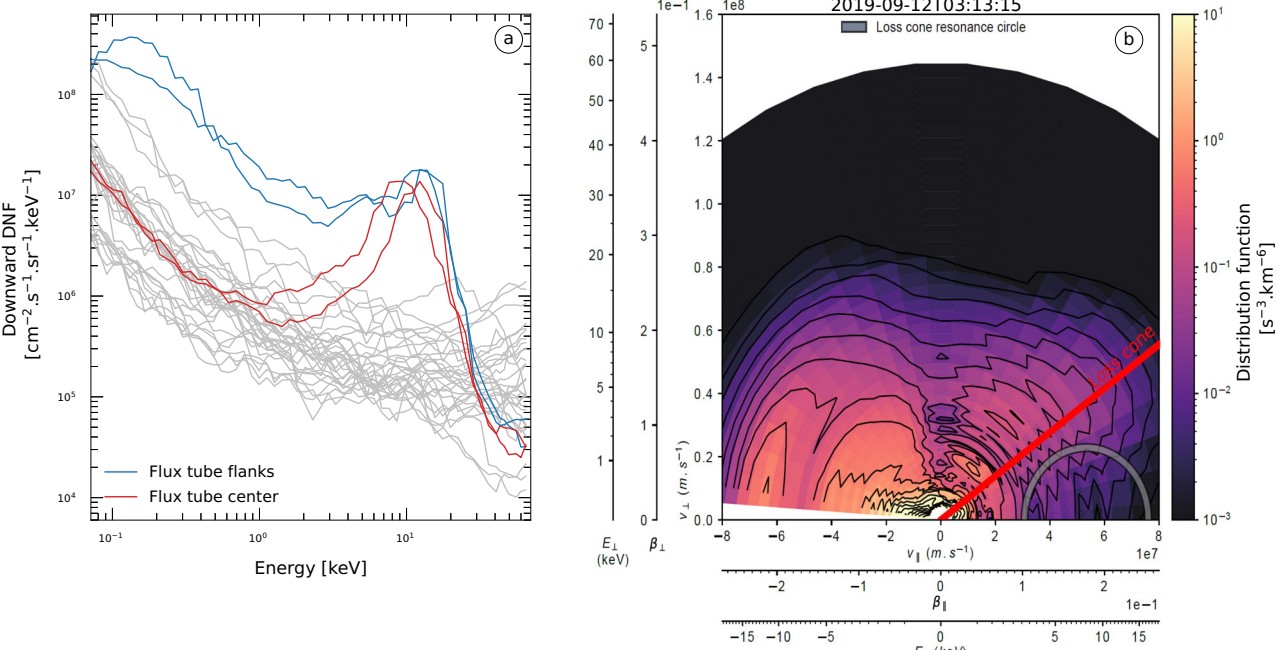

**Fig. 4 | Electron energy and velocity distributions within the Callisto flux tube.** **a** Electron energy distribution. Colored curves represent the distribution measured with a 1 s resolution during the flux tube crossing. The gray curves highlight the electron distributions measured during 30 s prior to the crossing, outside of the Callisto flux tube. **b** Electron distribution function in the velocity space ($v_{\parallel}$, $v_{\perp}$) at

the beginning of the flux tube crossing. The size of the loss cone is indicated by the red line. The gray circle indicates the resonance circle that maximizes the growth rate, enabling wave amplification. Note that at $v_{\parallel} = 0$ m.s$^{-1}$, the iso-energy contours decrease, which is due to spacecraft shadowing[88].

$5 \pm 1$ s of the flux tube crossing duration. Based on Juno's velocity during the flux tube crossing, i.e. about 50 km.s$^{-1}$, the source size is estimated to be $250 \pm 50$ km. The growth rates are maximum for electrons with characteristic energies ranging from 1 to 8 keV. The expected intensity of the waves is $0.6$ -$1.6 \times 10^{-18}$ W.m$^{-2}$.Hz$^{-1}$ (see "Cyclotron Maser Instability and growth rate calculation" in Methods), below the Juno-Waves current sensitivity, $5 \times 10^{-18}$ W.m$^{-2}$.Hz$^{-1}$. We therefore conclude that Callisto-induced radio emissions may exist, but that they would be too weak to be observed in situ during this perijove (Supplementary Fig. 6).

## Discussion and conclusions

The expansion of the main oval emission observed during Juno's PJ22 has enabled the Juno-UVS instrument to clearly detect a double-spot auroral structure along the Callisto footpath. The derivation of the rotation rate of these spots, as previously done to identify the Enceladus footprint at Saturn[50,51], in excellent agreement with the Callisto orbital velocity in the $S_{III}$ reference frame (0.0098 deg.s$^{-1}$), allows the unambiguous identification of this auroral structure as the Callisto footprint. This double-spot structure is associated with TEB and MAW spots, as previously observed in the footprints of the other Galilean moons. The UV brightness derived for these spots, i.e., $108 \pm 11$ and $137 \pm 15$ kR, are well below the brightness of the Io, Europa, and Ganymede footprints, confirming previous theoretical predictions[27]. These Juno remote observations confirm that under particularly expanded magnetospheric conditions, leading to an equatorward shift of the main auroral oval, observation of the Callisto footprint becomes possible. It is also suggested that in some limited longitude sectors the Callisto footpath appears to depart from the main oval, even in normal conditions (Supplementary Fig. 4), so that a reanalysis of the HST archive could provide more detections of the Callisto auroral footprint.

An estimate of the physical parameters in the current sheet ($H = 0.94 R_J$, $n_0^{electrons} = 0.15 \pm 0.02 cm^{-3}$) and at Callisto ($n_{at\ Callisto}^{electrons} =$

$0.1 \pm 0.01 cm^{-3}$) at the time of the observations has been derived from the longitudes of the spots, and are consistent with an expanded magnetosphere state of the Jovian system. This method could further be generalized to Io, Europa, and Ganymede using observations of their TEB and MAW auroral spots to remotely monitor the properties of the current sheet, e.g. ion and electron density or plasma scale height[69].

Magnetic mapping of the Juno position onto Jupiter's upper atmosphere shows that Juno crossed magnetic field lines connected to one spot of the Callisto footprint. This rare opportunity has been used to derive an integrated set of local physical parameters and characterize the moon-planet electrodynamic interaction responsible for this auroral emission, attributed to the TEB. Analysis of the electron measurements during this crossing revealed a downward field-aligned electron beam, with a 10 keV characteristic energy which can account for the excitation of the UV spot. The energy flux associated with the propagation of this electron beam is estimated at 55 mW.m$^{-2}$. In the meantime, a sharp enhancement in the wave spectral density demonstrates the presence of quasi-electrostatic waves, usually associated with high energy electron beams. In Table 1, we summarize the observations made during crossings of magnetic field lines connected to the TEB spots of Europa, Ganymede, and Callisto. For each of them, the electron energy distribution is non-monotonic, with energy in the keV range (4 - 18 keV), with a much stronger precipitating Electron Flux for Ganymede (316 mW.m$^{-2}$) than for Callisto (55 mW.m$^{-2}$) or Europa (36 mW.m$^{-2}$). These differences in electron energy between each moon can be explained by a difference in the initial power generated during the local moon-magnetosphere interaction, a difference in the efficiency of energy transfer from the moon to the acceleration regions, and/or a difference in the efficiency of wave-electron energy transfer. The in situ electron properties inducing Io's TEB spot have not been documented to date, although observations of this structure by the Hubble Space Telescope suggest that it is triggered by electrons whose population is

**Table 1 | Comparison of Juno observations obtained during Galilean moons' TEB flux tube crossings**

| | Europa[11,17,20] | Ganymede[11,18,21] | Callisto |
|---|---|---|---|
| In situ observation context | | | |
| Perijove | 12 N | 30 S | 22 N |
| Date | 2018-04-01 | 2020-11-08 | 2019-09-12 |
| Associated auroral structure | TEB | TEB | TEB |
| Electron properties | | | |
| EF [mW.m$^{-2}$] | 36 | 316 | 55 |
| EC [keV] | 3.6 | 18 | 10 |
| Energy distribution | Non-monotonic | Non-monotonic | Non-monotonic |
| Radio properties | | | |
| $f_{min}$ [MHz] | 6.7 | 1.8 | 7.75 |
| Intensity max [W.m$^{-2}$.Hz$^{-1}$] | $2.4 \times 10^{-7}$ | $7.2 \times 10^{-9}$ | $> 10^{-18}$ $< 5 \times 10^{-18}$ |
| UV properties | | | |
| Brightness [kR] | 37 | 411 ± 42 | 108 ± 11 |

Properties of the precipitating electrons, i.e. energy flux (EF), characteristic energy (EC), and type of energy distributions are indicated. We also provide the frequency and the intensity of the moon-induced radio emissions, as well as the UV brightness of the auroral footprints. References corresponding to the derived parameters are indicated in the column headings.

depleted below a few keV, suggesting a non-monotonic energy distribution[70].

The waves growth rates calculated during the Callisto TEB crossing and associated with loss cone instabilities in the electron distribution functions of the beams measured by Juno-JADE-E indicate that radio emissions can be triggered as part of the moon-planet interaction. However, their estimated intensity is below the Juno-Waves sensitivity threshold ($5 \times 10^{-18}$ W.m$^{-2}$.Hz$^{-1}$), due to a small wave growth rate and they were not detected. This also happens in some cases for other Galilean moons. Due to the short duration of the flux tube crossing, i.e. < 5 s, we did not analyze Juno-MAG data to derive the electric current properties, which generally requires smoothing of short time-scale variations.

These observations confirm the electrodynamic coupling between Callisto and Jupiter. This coupling will be further analyzed by the JUICE mission[71], successfully launched in April 2023, which will repeatedly visit Callisto and its local environment, enabling a better characterization of the interaction of Callisto with Jupiter's magnetosphere. The reported in situ and remote observations complete the family portrait of Galilean moon auroral footprints, and resolve the longstanding mystery as to whether Callisto's electromagnetic interaction is fundamentally different from the inner three Galilean satellites. The observed similarities, both in the auroral structure and the electron in situ properties, point towards a universal physical mechanism at work for moon-planet and planet-star magnetospheric interactions, relevant for other binary systems that are only accessible remotely, in the solar system and beyond.

# Methods

## UV maps
The complete mapping of the UV auroral emission above the poles of Jupiter is achieved by co-adding consecutive Juno-UVS measurements obtained between 68 and 210 nm. Each measurement consists of a 30-s scan of the auroral emissions, which corresponds to one spin of the Juno spacecraft. False color maps highlight the photon number recorded between 145 and 165 nm, which is a diagnostic of the depth of the auroral emission and the electron characteristic energy. This results from the fact that photons are preferentially absorbed at wavelengths shorter than 140 nm by methane, located deeper in the Jovian atmosphere. The UV photons measured by Juno-UVS are mapped onto Jupiter's latitude/longitude grid, assuming they are emitted at 900 km-altitude above Jupiter's 1 bar-level, which corresponds to the mean altitude of the moon-UV induced aurora[72]. The UV brightnesses were calculated using the method presented by ref. 73. This consists first in integrating the photons recorded between 115–118 nm and 125–165 nm. The latter is then multiplied by 1.82 to extrapolate the brightness over the total H$_2$ and Lyman-$\alpha$ emissions, i.e. in the 75–198 nm range, using a H$_2$ synthetic spectrum from ref. 74. This synthetic spectrum was simulated by accounting for the Lyman, Werner and Rydberg band systems of H$_2$, assuming 300 K for the rotational and vibration H$_2$ temperatures, and excited by a mono-energetic electron beam of 100 eV. The spectrum was simulated as non-absorbed by the Jovian stratospheric hydrocarbons, and self-absorption was not accounted for.

## Rotation rate
Consecutive spin-by-spin images from the Juno-UVS instrument are used to derive the rotation rate of the identified spots. In each frame, the spot longitude $\phi_{SIII}$ and latitude $\lambda_{SIII}$ are retrieved using a polar grid superimposed on the UVS observations. The location of the spot is then magnetically back-traced onto the orbital plane of Callisto. By investigating consecutive measurements, the equatorial longitude evolution of the spot as a function of time can be deduced and compared with Callisto's angular velocity in the S$_{III}$ reference frame. This latter is found using Callisto's synodic period $P_{Syn}^{Cal} = (P_{Jup} \times P_{Cal})/(P_{Cal} - P_{Jup}) = 10.177 \, h$ and then derived using $d\lambda/dt = 360°/P_{syn}^{Cal} = 0.0098 \, deg.s^{-1}$.

## Estimate of the electron density at Callisto
Using the longitude separation of the Callisto TEB and MAW spots identified and simple geometric considerations, an estimate of the plasma density near Callisto can be made.

We first demonstrate that if there is no plasma dense enough to slow down the propagation of Alfvén waves, the TEB and MAW spots (if existing) should almost be superimposed. More precisely, the equatorial longitude separation $\Delta\lambda$ is calculated as:

$$\Delta\lambda = \Delta t \times \omega_{Callisto} \qquad (1)$$

with $\omega_{Callisto} = 0.0098 \, deg.s^{-1}$ the rotation rate of Callisto, and $\Delta t = t_{TEB} - t_{MAW}$. The travel times $t_{TEB}$ and $t_{MAW}$ associated with the propagation of Alfvén waves between Callisto and the TEB and MAW spots in the northern hemisphere are given by:

$$t_{TEB} = \frac{L_S}{c} + \frac{L_S + L_N}{v} \, and \, t_{MAW} = \frac{L_N}{c} \qquad (2)$$

with $L_N$ and $L_S$ the length of the magnetic field lines from Callisto to Jupiter in the Northern and Southern hemisphere, respectively, $c$ the speed of light in vacuum and $v = \sqrt{\frac{2E}{m_e}}$ the classical electron speed. The length of the magnetic field lines are determined using the JRM33 + KK2005 magnetic field model (see ref. 75 and "Magnetic mapping" in Methods).

In the present case, Callisto is located at jovigraphic latitude $\theta_{S_{III}} = -0.24°$ and at an equatorial longitude $\lambda_{III} = 306.93°$. The characteristic energy of the electron is taken as $E = m_e v^2 / 2 = 10$ keV. Therefore, the equatorial longitudes for northern TEB and MAW spots should be $\lambda_{TEB} = 306.17°$ and $\lambda_{MAW} = 306.86°$, respectively. Therefore, the separation between the MAW and TEB spots is $\Delta\lambda = 0.69°$ in the case where there is no plasma in the equatorial plane of the magnetosphere.

Now considering that there is a plasma dense enough to slow down the propagation of Alfvén waves, then inside this plasma sheet

**Table 2 | Estimated plasma properties close to Callisto and at the centrifugal equator during PJ22**

| Parameters | Plasma scale height | | | | | |
| --- | --- | --- | --- | --- | --- | --- |
| | H = 0.75 R$_J$ | H = 0.94 R$_J$ | H = 1.00 R$_J$ | H = 1.50 R$_J$ | H = 2.00 R$_J$ | H = 3.00 R$_J$ |
| $\rho_0$ (× 10$^{-27}$ kg.cm$^{-3}$) | 3.87 ± 0.51 | 3.63 ± 0.48 | 3.58 ± 0.47 | 3.36 ± 0.44 | 3.27 ± 0.43 | 3.16 ± 0.41 |
| $\rho_{at\,Callisto}$ (× 10$^{-27}$ kg.cm$^{-3}$) | 2.20 ± 0.29 | 2.31 ± 0.30 | 2.35 ± 0.31 | 2.53 ± 0.33 | 2.64 ± 0.34 | 2.75 ± 0.36 |
| $n_0^{electrons}$ (cm$^{-3}$) | 0.160 ± 0.022 | 0.150 ± 0.020 | 0.147 ± 0.019 | 0.138 ± 0.018 | 0.135 ± 0.018 | 0.130 ± 0.017 |
| $n_{at\,Callisto}^{electrons}$ (cm$^{-3}$) | 0.091 ± 0.013 | 0.095 ± 0.012 | 0.097 ± 0.013 | 0.105 ± 0.014 | 0.109 ± 0.014 | 0.113 ± 0.015 |
| $\lambda_{MAW}$ (°) | 298.62 | 297.70 | 297.44 | 295.56 | 294.21 | 292.39 |
| $\lambda_{TEB}$ (°) | 303.97 | 303.05 | 302.79 | 300.91 | 299.56 | 297.73 |

Total plasma density at the center of the current sheet $\rho_0$, at Callisto $\rho_{at\,Callisto}$, electron density at the center of the current sheet $n_0^{electrons}$, and electron density at Callisto $n_{at\,Callisto}^{electrons}$. The estimated equatorial longitudes of the MAW and TEB spots, for each plasma sheet scale height $H$ and for an electron characteristic energy $E = 10\ keV$ are indicated in the last two rows. The best fit correpond to H = 0.94 R$_J$ column.

(PS) the group velocity $v_A$ of the Alfvén waves is:

$$v_A = \frac{B}{\sqrt{\mu_0 \rho}} \qquad (3)$$

with $B$ the local magnetic field strength, $\rho$ the local plasma mass density, and $\mu_0$ the vacuum permeability. Outside of the plasma sheet, where the plasma density is low, we consider the Alfvén speed to be the speed of light $c$. Therefore, the travel time t$_{TEB}$ and t$_{MAW}$ associated with the propagation of the Alfvén waves from Callisto to the northern spots are now:

$$t_{TEB} = \frac{L_{Sin\,PS}}{v_A} + \frac{L_{Sout\,PS}}{c} + \frac{L_S + L_N}{v} \; and \; t_{MAW} = \frac{L_{N\,in\,PS}}{v_A} + \frac{L_{N\,out\,PS}}{c} \qquad (4)$$

with L$_{S\,in\,PS}$, L$_{N\,in\,PS}$ the length of the northward and southward magnetic field lines in the plasma sheet, L$_{S\,out\,PS}$, L$_{N\,out\,PS}$ the length of the northward and southward magnetic field lines outside of the plasma sheet. L$_N$ = L$_{N\,in\,PS}$ + L$_{N\,out\,PS}$ and L$_S$ = L$_{S\,in\,PS}$ + L$_{S\,out\,PS}$ represent the total length of the magnetic field lines from Callisto to Jupiter's northern and southern auroral regions, respectively. Note that a tilt $\theta$ of the Alfvén wings, depending on the Mach Alfvén number $M_A$ as $\theta = atan(M_A)$, exists and increases $L_{S\,in\,PS}$ and $L_{N\,in\,PS}$. $M_A$ depends on the Alfvén speed $v_A$ and therefore on the plasma density $\rho$. An uncertainty factor $\Delta L$ will therefore be taken on the lengths $L_{S\,in\,PS}$ and $L_{N\,in\,PS}$, which will be propagated to calculate an uncertainty on the value of $\rho$.

The major constraints are the position of the UV footprint and their associated equatorial longitudes $\lambda_{III}$. In the present study, $\lambda_{TEB\,observed} = 303.05°$, $\lambda_{MAW\,observed} = 297.7°$, therefore $\Delta\lambda_{observed} = 5.35°$.

Using these constraints and the previous formalism, we estimate the plasma mass density as follows:

1. We first assume the PS to be aligned with the centrifugal equator, i.e., at $\theta = 3.1°$ (in the $\lambda_{III} = 204.2°$ direction) of the jovicentric equator. We then assume a plasma sheet height scale $H$, which gives the values of $L_{S\,in\,PS}$, $L_{S\,out\,PS}$, $L_{N\,in\,PS}$ and $L_{N\,out\,PS}$.
2. We assume a plasma density $\rho_0$ at the center of the PS and we determine $\rho_i$ the density along the magnetic field line in the PS using the following density profile equation[76]:

$$\rho_i = \rho_0 \, exp\left(-\sqrt{\frac{(r_i - r_0)^2 + z_i^2}{H}}\right) \qquad (5)$$

with $r_0$ the equatorial diameter set to Callisto's orbital distance, i.e., $r_0 = 26.33\,R_J$, $r_i = \sqrt{x_i^2 + y_i^2}$ the equatorial radial distance, $z_i$ the altitude above the equator of the position of the measurement point $i$, and $H$ the plasma sheet scale height.

3. We determine the Alfvén speed velocity $v_A$, based on the calculated $\rho_i$ and the magnetic field amplitude $B_i$ using the JRM33 + KK2005 magnetic field model. From that, we obtain the values of $t_{TEB}, t_{MAW}, \lambda_{TEB}, \lambda_{MAW}$ and therefore $\Delta\lambda_{calculated}$.
4. By applying this method on different values of $\rho_0$, we minimize $|\Delta\lambda_{calculated} - \Delta\lambda_{observed}|$.
5. We then run the same above calculations for different values of the scale height $H$ to minimize $|\lambda_{MAW\,calculated} - \lambda_{MAW\,observed}|$ and $|\lambda_{TEB\,calculated} - \lambda_{TEB\,observed}|$.

Table 2 summarizes the results. The best result gives MAW and TEB footprint equatorial longitudes $\lambda_{MAW\,calculated} = 297.70°$ and $\lambda_{TEB\,calculated} = 303.05°$. This is obtained for a plasma sheet scale height $H = 0.94\,R_J$, and a density $\rho_0 = 3.6 \pm 0.5 \times 10^{-27}$ kg.cm$^{-3}$, which corresponds to a density at Callisto $\rho_{at\,Callisto} = 2.3 \pm 0.3 \times 10^{-27}$ kg.cm$^{-3}$. Note that the uncertainties are based on the tilt angle of the Alfvén wings, which increases the length of the magnetic field line in the plasma sheet as $L = L_{in\,PS} \times \Delta L$ with $\Delta L = 1/\cos\theta = 1.14$, with $\theta = atan(M_A)$ for $M_A = 0.55$. We therefore take the mean value between the results with $\Delta L = 1$ and $\Delta L = 1.14$

Using the plasma distribution model of ref. 77, we estimate that at Callisto's orbital distance and at the centrifugal equator, the plasma consists of 54.7% sulfur ions (9.9% S$^+$, 35.1% S$^{++}$, 9.7% S$^{+++}$), 26.6% oxygen ions (21.6% O$^+$, 5% O$^{++}$), 12.5 % protons H$^+$, and 6.2% sodium ions Na$^+$. Given this composition, we calculate that the average ion mass $m_{mean}$ is 23.3 amu (3.87 × 10$^{-26}$ kg) and the average ion charge $q_{mean}$ is $+ 1.6q_e$. Based on these values, we estimate that the ion density at the center of the current sheet and at Callisto, given by $n^{ions} = \rho/m_{mean}$, are $n_0^{ions} = 0.094 \pm 0.012$ cm$^{-3}$ and $n_{at\,Callisto}^{ions} = 0.060 \pm 0.008$ cm$^{-3}$, respectively. We derive the associated electron densities using quasi-neutrality assumption, i.e., $n^{electrons} = q_{mean}n^{ions}$, leading to $n_0^{electrons} = 0.15 \pm 0.02$ cm$^{-3}$ and $n_{at\,Callisto}^{electrons} = 0.095 \pm 0.012$ cm$^{-3}$.

## Electron energy flux and characteristic energy

The downward electron energy flux (mW.m$^{-2}$), i.e., the energy flux precipitating into Jupiter's atmosphere and inducing aurora, is estimated from Juno-JADE-E measurements of electrons within the loss cone. The size of the loss cone at the measurement time is estimated by $\sin^{-1}(r^{-3/2})$ where r is the distance from the Juno spacecraft to the center of Jupiter. Electron differential number flux (DNF, [cm$^{-2}$.s$^{-1}$.sr$^{-1}$.keV$^{-1}$]) within the loss cone is then converted into energy flux, EF, by:

$$EF = \pi \sum_{E\,min}^{E\,max} DNF \times E \times \Delta E \qquad (6)$$

where the summation is performed on the JADE-E energy channels, with E and $\Delta E$ refer to the geometric mean value and the energy width of each energy channel, respectively. The characteristic energy of the

downward electrons, EC, is derived using:

$$EC = \frac{\sum_{E\min}^{E\max} DNF \times E \times \Delta E}{\sum_{E\min}^{E\max} DNF \times \Delta E} \quad (7)$$

with E and $\Delta E$ the geometric mean value and the energy width of each energy channel, respectively.

## Magnetic mapping

The footpaths of the Galilean moons and the Juno spacecraft are derived by an iterative follow-up of the magnetic field line, with a constant step size of $1/300\, R_J \simeq 240$ km between the object of interest and Jupiter's atmosphere. The footpaths are computed at a 900-km altitude above the 1-bar level, corresponding to the mean altitude of the moon-UV induced aurora[78]. To ensure that the magnetic field line mapping is as accurate as possible, we use two different models depending on the radial distance of the object to be mapped and the direction of the field line tracing. To derive the footpaths of Io, Europa, Ganymede, and Juno above the auroral regions of Jupiter, i.e. for M < 20, we use the JRM33 + CON2020 model, a combination of an intrinsic and external magnetic field model based on Juno-MAG data. The footpath of Callisto and Juno M-Shell for M > 20 are inferred with the JRM33 + KK2005 model, as it gives more accurate estimates of the magnetic field components near the orbit of Callisto (Supplementary Fig. 7), in the middle, and outer magnetosphere, i.e., r > 20 $R_J$[75]. The moons' footpaths are derived by considering only magnetic field models. Consequently, no effect of the propagation time of the waves and particles between the moons and Jupiter's atmosphere are taken into account in this calculation.

The M-Shell parameter is defined as the distance between Jupiter's center and the minimum of the magnetic field strength along the field line. This latter is computed by an iterative tracing of the magnetic field lines until the minimum of the magnetic field strength is reached.

We emphasize that the use of the JRM33 (13th-order) model to describe Jupiter's internal magnetic field constitutes a major step forward in the unambiguous identification of the Callisto footprint reported in this study. Indeed, such a model allows much more precise and detailed estimates of Jupiter's magnetic field than was previously possible with the VIP4 (4th-order) internal magnetic field model[79] used in the previous tentative detections of the Callisto auroral footprint.

## Cyclotron Maser Instability and growth rate calculation

Amplification of radio waves can occur through the Cyclotron Maser Instability (CMI) under different conditions: **(i)** the plasma needs to be tenuous and magnetized to fulfill $f_{pe} \ll f_{ce}$ with $f_{pe} = \frac{1}{2\pi}\left(\frac{n_e q^2}{\epsilon_0 m_e}\right)^{0.5}$ the electron plasma frequency and $f_{ce} = \frac{1}{2\pi}\left(\frac{qB}{m_e}\right)$ the electron cyclotron frequency, **(ii)** the presence of hot, weakly relativistic and unstable electrons generally embedded within a cold, prominent, electron population. The CMI amplifies waves near the electron cyclotron gyrofrequency $\omega_{ce} = 2\pi f_{ce}$ along the resonance equation $\omega = \frac{\omega_{ce}}{\Gamma} + k_{\parallel} v_{\parallel}$

where $\omega = 2\pi f$ is the wave angular frequency, $\Gamma^{-1} = \sqrt{1 - \frac{v^2}{c^2}}$ is the Lorentz factor and $k_{\parallel}$ and $v_{\parallel}$ are the projection of the wave vector **k** and the electron velocity **v** onto the direction of the local magnetic field. In the ($v_{\perp}$, $v_{\parallel}$) phase space, the resonance equation transposes into the equation of a circle defined by its center $v_0 = \frac{k_{\parallel} c^2}{\omega_{ce}}$ and its radius $v_r = \sqrt{v^2 - 2c^2\Delta\omega}$ with $\Delta\omega = (\omega - \omega_{ce})/\omega_{ce}$.

Waves are amplified whenever the wave growth rate computed from the EDF $F$ along the resonance circle is positive. The analytical expression of the growth rate results from the right-handed extraordinary (RX) mode dispersion equation. The latter depends on the plasma properties[11,80]:

$$\gamma = \frac{\left(\frac{\pi}{2}\epsilon_h\right)^2}{1 + \left(\frac{\epsilon_c}{2\Delta\omega}\right)^2} c^2 \int_0^\pi d\theta\, v_r^2 \sin^2(\theta)\frac{\partial F_h}{\partial v_\perp}(v_0 + v_r\cos(\theta), v_r\sin(\theta)) \quad (8)$$

where $F_h$ represents the normalized electron distribution, $\epsilon_h = \frac{\omega_{ph}}{\omega_{ce}}$ and $\epsilon_c = \frac{\omega_{pc}}{\omega_{ce}}$ with $\omega_{ph}$, $\omega_{pc}$ the plasma frequency of the hot and cold electrons, respectively.

This equation means that the CMI free energy source lies in the EDF portion where $\frac{\partial F_h}{\partial v_\perp}$ is positive. The growth rate is the integral of the perpendicular gradient of the hot EDF $\frac{\partial F_h}{\partial v_\perp}$ along the CMI resonance circle in the velocity space.

To derive the expected intensity of the amplified wave $S_{Radio}$, we supposed a homogeneous source of latitudinal extent $L_C$ with a constant growth rate $\gamma$. We assumed that the CMI mechanism amplifies galactic noise of intensity $S_{Source}$ ($10^{-19} W.m^{-2}.Hz^{-1}$ at 10 MHz[81]). The gain $\frac{S_{Radio}}{S_{Source}}$ is then given by: $\frac{S_{Radio}}{S_{Source}} = \exp(\frac{4\pi f_{ce}\gamma L_C}{v_g})$.

For the sake of simplicity, we used the value of the group velocity $v_g = 0.1c$[82].

## Data availability

The JADE-E (https://pds-ppi.igpp.ucla.edu/collection/JNO-J_SW-JAD-3-CALIBRATED-V1.0), Waves (https://pds-ppi.igpp.ucla.edu/collection/JNO-E_J_SS-WAV-3-CDR-BSTFULL-V2.0), UVS and MAG (https://pds-ppi.igpp.ucla.edu/collection/JNO-J-3-FGM-CAL-V1.0) datasets used in this study are publicly available in the Planetary Data System (PDS) database (https://pds-ppi.igpp.ucla.edu/). The $H_2$ synthetic spectrum used to calculate the UV brightness is provided in Supplementary Fig. 8. Hubble Space Telescope observations presented in Supplementary Fig. 1 were retrieved from the APIS database (APIS)[83]. Source data are provided with this paper. Data generated in this study are available at https://doi.org/10.5281/zenodo.15423510 Source data are provided with this paper.

## Code availability

The JRM33 + KK2005 model used to compute the Callisto footpath can be downloaded at https://zenodo.org/records/10102742. The JRM33 + CON2020 model is publicly available at https://github.com/mattkjames7/JupiterMag. Data analysis of the Solar Wind characteristics derived from Tao et al. (2005) was performed with the AMDA (AMDA)[84] science analysis system provided by the Centre de Données de la Physique des Plasmas (CDPP) supported by CNRS, CNES, Observatoire de Paris and Université Paul Sabatier, Toulouse. The numerical model[85] used to estimate the equatorial density and the plasma sheet height from Juno-UVS observations of the TEB and MAW auroral spots can be downloaded at https://doi.org/10.5281/zenodo.1534817081.

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

## Acknowledgements

French co-authors acknowledge the support of CNES for the Juno and JUICE missions. This study has been partially supported through the grant EUR TESS N°ANR-18-EURE-0018 in the framework of the Programme des Investissements d'Avenir. V. H. acknowledges support from the French government under the France 2030 investment plan, as part of the Initiative d'Excellence d'Aix-Marseille Université – A*MIDEX AMX-22-CPJ-04. French authors acknowledge the support of CNRS/INSU national programs of planetology (PNP) and heliophysics (PNST). The work at SwRI was funded by the NASA New Frontiers Program for Juno through contract NNM06AA75C. B. B. is a Research Associate of the Fonds de la Recherche Scientifique, FNRS. We thank Jacques Gustin for providing us with a synthetic $H_2$ spectrum of the Jovian auroras.

## Author contributions

J.R. conceived the study, analyzed the JADE data, and wrote the manuscript. V.H. analyzed the UVS data. C.L. and B.C. analyzed the Waves data. N.A. and Q.N. helped with the initial draft of the manuscript. J.S, F.A., and R.W.E., P.L, and E.P. contributed to the JADE data analysis. E.P. is responsible for the JADE data distribution at IRAP. T.K.G., G.R.G., B.B., R.S.G., J.A.K., M.H.V. contributed to the UVS data analysis. A.H.S., W.S.K contributed to the Waves data analysis. L.L. and P.Z. contributed to the validation and interpretation of the results. D.S.C and A.K. contributed to the conception of the study. C.K.L., R.P., and J.R. conceived the method to infer the equatorial plasma density from the Juno-UVS observations. J.E.P.C. is the Principal Investigator of the MAG instrument. S.J.B. is the Principal Investigator of the Juno mission. All authors contributed to the discussion of results, read and provided feedback on the manuscript.

## Competing interests

The authors declare no competing interests.

## Additional information

J. Rabia.

