## [Transparent Peer Review file · Nature Communications]

Juno in situ and remote observations of the ultraviolet footprint of the moon Callisto

Corresponding Author: Mr Jonas Rabia

Version 0:

Reviewer comments:

Reviewer #1

(Remarks to the Author)

Summary: This manuscript presents evidence of the detection of Callisto's auroral footprint using Juno observations of Jupiter in the UV. The rotation rate of the footprint is found to align with the movement of Callisto in its orbit around Jupiter. The UV evidence is further corroborated by the Juno-JADE observations of electron flux as the spacecraft flew through the magnetic flux tube connecting Callisto with Jupiter. The authors present a thorough and well-organized analysis of the data and the evidence for the detection of the footprint of Callisto in the Juno data. My comments on the manuscript are minor and are as follows:

1. Line 71: "However, improvements in the Jovian magnetic field modeling made these previous observations questionable." The authors should describe what improvements in the magnetic field model were made and how was the previous model used in the HST analysis is lacking compared to this model. The section on "Magnetic Mapping" would be such a place to present this discussion.
2. Line 84-85: "Here, we take advantage of unusual magnetospheric conditions..." This sentence is written in a manner that makes it seem like this is the first time the main auroral oval at Jupiter was shifted which allowed the authors an opportunity to find evidence of Callisto's footprint, which is otherwise buried in the much brighter main oval. The Bhattacharyya et al., 2018 study has also pointed out this effect. The present study fails to recognize that fact. Adding "... similar to the previously reported detection with HST" or something like that at the end of the sentence would suffice. The same is true for the sentence presented in line 314. "These Juno remote observations establish that ..." Such an occurrence has already been established through HST observations. The Juno observations is not the first to establish this fact that the Callisto footprint is likely to be visible when the main oval moves equatorward.
3. Line 148: Mentions a pair of spots for Io and Europa as observed with Juno. But fig. 1b identifies two spots for Ganymede and Callisto, but only one for Io and Europa. Please rectify/or explain this mismatch in the text.
4. Line 152: Why do certain images presented in the time sequence of Fig. 2 do not have the secondary spot for Callisto? Please elaborate.
5. Line 201: Please justify quantitatively the words "appears redder" in the sentence.
6. Line 226: Please use a different word than "footpath". I am not sure what the authors mean by this term. It does appear in many places in the text. Please rectify everything.
7. Line 282: "This strikingly compares to the source EDF for decametric emission by Io, Europa, and Ganymede." Please elaborate on what the authors mean by the phrase "strikingly compares".
8. Line 324: Please add "... expanded magnetospheric state of the Jovian system."
9. Line 324: "This method could further be generalized to monitor..." What method are the authors referring to here. Please elaborate.
10. Line 365: Please rewrite "...Callisto-magnetosphere interactions" to "...interactions of Callisto with Jupiter's magnetosphere."

Reviewer #2

(Remarks to the Author)

The paper addresses an important issue of electromagnetic sub-Alfvénic planet-satellite interactions, which is relevant for planetary and exoplanetary systems.

The authors describe in detail the Juno observations and explain the uniqueness and difficulty of finding the temporarily existing auroral emission from Callisto, which requires special conditions in the planetary magnetosphere. They describe the specific conditions that made such observation possible. These conditions were reduced to a combination of two circumstances: the displacement of the main oval of Jupiter's polar auroras toward the equator due to the reduced dynamic pressure of the solar wind and the localization of Callisto in the sub-Alfvénic zone due to the displacement of its orbit from the plasma sheet to a region with decreased plasma density and increased magnetic field. The authors emphasize the correlation between the expansions of the magnetosphere and the main oval.

Usually, the footprint of Callisto is not observed in Jupiter's auroral pattern, unlike the other three Galilean moons: Io, Europa and Ganymede. When Callisto is embedded in the plasma sheet, it is outside the Alfvén radius, in the super-Alfvénic flow, and generates a comet-like magnetosphere, that is not directly linked with Jupiter's aurora. Even when magnetospheric conditions near Callisto change so that the plasma flow becomes sub-Alfvénic, Callisto's footprint is weak and located within (or nearby) the bright main auroral oval.

The data analyzed are obtained with appropriate techniques and interpreted carefully.

The paper can be published after minor revision.

To make the content of the article understandable to a reader who is not immersed in this topic, it is advisable to make the following clarifications.

1. It would make sense to remind the reader why the violation of solid corotation of magnetospheric plasma occurs at a distance of $\sim 20\text{-}40$ RJ from the center of Jupiter (what is the reason) and why the footprints from the 3 Galilean satellites (Io, Europa and Ganymede) are located equatorward from the main auroral oval, while the Callisto's one is poleward from it.

2 It is desirable to explain more clearly why the "increased outflow of plasma mass from the Io torus" is due to the expansion of the magnetosphere and why it leads to a shift of the main auroral oval equatorward? Now the reader is only referred to the specified references.

3. It is desirable to mention what kind of magnetospheric structure arises around Callisto when it is in the super-Alfvénic flow of Jupiter's magnetospheric plasma, which will make it impossible for this moon to generate auroras at the planet.

4. It would be worthwhile to more clearly indicate what is the source of energy for generating the auroras from Callisto and the other Galilean satellites.

Reviewer #3

(Remarks to the Author)

This paper reports results of JUNO observations that take advantage of an expansion of the Jovian magnetosphere to positively detect the auroral footprint of the most distant Galilean satellite, Callisto, and to perform a detailed multispectral comparative study with respect to the other three Galilean satellites that were also included in the JUNO observations. The footprint represents a Jovian aurora created by a dynamo-driven electric current traveling along the magnetic flux tube connecting a satellite to Jupiter's upper atmosphere. The footprint of Callisto is not normally visible because it is dimmer and lies closest to the poles and so is easily lost in Jupiter's bright polar aurora. The footprints of the other three satellites have previously been observed and studied, revealing information about the magnetosphere and the role that the Jovian satellites play in Jupiter's aurorae.

This study probes the magnetosphere at higher magnetic latitudes, as well as providing new information about the plasma sheet at the magnetic equator, near the plane of the satellite orbits, where the electron current originates. The reported analysis benefits from the use of in situ measurements by the JUNO instruments within the magnetic flux tubes connecting the satellite to the auroral footprint. These yield quantitative results, for the electron flux, characteristic electron energy, deposited auroral energy flux, and the thickness of the equatorial current sheet.

The results will be of interest to a wide community of plasma scientists interested in planetary magnetospheres. They will also bear on other giant planets having a magnetosphere.

The paper is well-written with abundant references and clarity, which is important because of the multiple magnetospheric processes at play and the many variables affecting the phenomena. The Figures are of high quality and the Tables convey relevant information. I recommend this paper for publication in Nature Communications.

The following lists several minor comments that should be addressed:

Fig. 2a shows a background white dotted grid that is not explained. Presumably, it belongs to Jupiter's body frame since it shows Callisto's footpath cutting ~ 45 degrees across the high-latitude grid instead of being quasi-meridional. This could result from the tilted magnetic pole, which is the frame for the footpath. Does Juno's relative motion, or the speed of the Alfvén wave (Extended Fig. 5) affect this footpath angle?

Adding an explanatory comment to the caption or text would be beneficial.

Extended Fig. 5: Is there a change in direction of the footpath where the propagation speed changes at the boundary of the high density equatorial region? Or is the footpath constrained at all propagation speeds by the tension in the magnetic flux line?

Line 260: "of in" Choose which word to use.

Line 309: Need a comma after the closing parenthesis.

Line 490: Define "CS" (current sheet?)

Line 533: f_{pe} is undefined (electron plasma frequency?)

Line 544: RX is undefined (Radio frequency?)

Line 549: Where is f_h used?

Laurence Trafton

Version 1:

Reviewer comments:

Reviewer #1

(Remarks to the Author)

I am satisfied with the author responses to my comments and the subsequent changes made to the manuscript in response to the comments. I have no additional comments on the manuscript.

Reviewer #2

(Remarks to the Author)

Reviewer #2

1. It would make sense to remind the reader why the violation of solid corotation of magnetospheric plasma occurs at a distance of ~ 20 -40 RJ from the center of Jupiter (what is the reason) and why the footprints from the 3 Galilean satellites (Io, Europa and Ganymede) are located equatorward from the main auroral oval, while the Callisto's one is poleward from it. \rightarrow We agree with the reviewer that explaining why the corotation breakdown and main oval map to ~ 20 -40 RJ is interesting. It is a balance of MIT coupling, strength of the internal magnetic field, effects of the current sheet field, and plasma properties. However, expanding on this topic is not mandatory in order to understand our analysis of the Callisto footprint. The central point is that the orbit of Callisto at $R=26$ RJ coincides with the mapping of the main oval. We therefore do not further describe the origin of corotation breakdown in the revised article.

\rightarrow We have nonetheless added an explanation of the location of the moons' auroral footprints with respect to the main oval : "Because the corotation breakdown is expected to occur at radial distances larger than the orbital distances of Io, Europa, and Ganymede, their UV and IR auroral footprints are located equatorward from the main auroral oval on Jupiter's polar regions."

Does this mean that Callisto is outside the corotation breakdown, as its auroral footprint is located poleward of the main oval? If so, the corotation breakdown distance must be between 20 and 26 RJ in the case under consideration. If not, it would be desirable to clarify this issue.

2 It is desirable to explain more clearly why the "increased outflow of plasma mass from the Io torus" is due to the expansion of the magnetosphere and why it leads to a shift of the main auroral oval equatorward? Now the reader is only referred to the specified references.

\rightarrow We have added a description of the processes leading to an equatorward shift of the main oval, introducing the stretching of the magnetic field line, as considered by theoretical models and observational studies.

3. It is desirable to mention what kind of magnetospheric structure arises around Callisto when it is in the super-Alfvénic flow of Jupiter's magnetospheric plasma, which will make it impossible for this moon to generate auroras at the planet.

\rightarrow We precised in the text that in the case of a slightly-super-Alfvénic interaction, as experienced by Callisto when located at the center of the plasma sheet, no clear bow shock structure is formed (e.g. Ridley 2007, doi:10.5194/angeo-25-533-2007, simulated the formation of Earth's bow shock formation at $MA \sim 8$). Still, even if a bow shock does not form, the energy transfer efficiency between the Callisto environment and Jupiter's polar region may be significantly reduced.

4. It would be worthwhile to more clearly indicate what is the source of energy for generating the auroras from Callisto and the other Galilean satellites.

\rightarrow As it was not clearly stated in the text, we have added the following sentence in the introduction "While UV and IR auroral emissions are induced by electron precipitation within Jupiter's atmosphere, moon-induced radio emissions result from unstable electron population propagating away from Jupiter."

And what is the source of energy for this process?

I'm not against the publication of this article.

Regards

Reviewer #3

(Remarks to the Author)

The authors have responded satisfactorily to my comments and have my approval for publication in Nature Communications.

Callisto's ultraviolet footprint: first simultaneous in situ and remote characterization using Juno data

J. Rabia, V. Hue, C.K. Louis, N. André, Q. Nénon, J.R. Szalay, R. Prangé, L. Lamy, P. Zarka, B. Collet, F. Allegrini, R.W. Ebert, T.K. Greathouse, B. Bonfond, G.R. Gladstone, A.H. Sulaiman, W.S. Kurth, J.E.P. Connerney, P. Louarn, E. Penou, A. Kamran, D. Santos-Costa, R.S. Giles, J.A. Kammer, M.H. Versteeg, and S.J. Bolton

Response to the reviewers

We thank the three reviewers for their careful review of our article and their constructive comments. We have included them in the manuscript. We respond hereafter to the reviewers' comments and we specify the changes made to the manuscript accordingly.

Reviewer #1:

1. Line 71: "However, improvements in the Jovian magnetic field modeling made these previous observations questionable." The authors should describe what improvements in the magnetic field model were made and how was the previous model used in the HST analysis is lacking compared to this model. The section on "Magnetic Mapping" would be such a place to present this discussion.

→ **The main improvement in the magnetic field model resides in the fact that the JRM33 model used here is a 13th-order model, compared to the VIP4 model used before which was a 4th order model. The magnetic field is therefore modelled with much more accuracy and refinement. As suggested by the reviewer, we have added a discussion of this improvement in the "Magnetic Mapping" section.**

2. Line 84-85: "Here, we take advantage of unusual magnetospheric conditions..." This sentence is written in a manner that makes it seem like this is the first time the main auroral oval at Jupiter was shifted which allowed the authors an opportunity to find evidence of Callisto's footprint, which is otherwise buried in the much brighter main oval. The Bhattacharyya et al., 2018 study has also pointed out this effect. The present study fails to

recognize that fact. Adding "... similar to the previously reported detection with HST" or something like that at the end of the sentence would suffice. The same is true for the sentence presented in line 314. "These Juno remote observations establish that ...". Such an occurrence has already been established through HST observations. The Juno observation is not the first to establish this fact that the Callisto footprint is likely to be visible when the main oval moves equatorward.

→ **We agree with the comment of the reviewer. We have added in the introduction the following sentence : "[...] and also suggested by the previous tentative detection of the Callisto footprint". We have also modified the sentence in the conclusion by "These Juno remote observations confirm that under particularly expanded magnetospheric conditions [...]"**

3. Line 148: Mentions a pair of spots for Io and Europa as observed with Juno. But fig. 1b identifies two spots for Ganymede and Callisto, but only one for Io and Europa. Please rectify/or explain this mismatch in the text.

→ **The original sentence referred to identification of the auroral tails of Io and Europa, but we understand that the sentence was misunderstood. We rephrased it and added an explanation on why only one spot is observed in the footprints of Io and Europa here.**

4. Line 152: Why do certain images presented in the time sequence of Fig. 2 do not have the secondary spot for Callisto? Please elaborate.

→ **Juno-UVS consists of a photon-counting imaging spectrograph that has a 7.2°-long slit. UVS also has a scan mirror, which allows shifting the instrument field of view to target different features of interest on Jupiter. Fig. 2 shows a sequence of almost consecutive spin-by-spin images. In this sequence, UVS is using several mirror pointings across Jupiter. This results in a partial coverage of the two Callisto auroral footprints, with sometimes only one of the two spots visible in the instrument field of view.**

5. Line 201: Please justify quantitatively the words "appears redder" in the sentence.

→ **We added a quantification by comparing the false color intensity with that of the polar emissions.**

6. Line 226: Please use a different word than "footpath". I am not sure what the authors mean by this term. It does appear in many places in the text. Please rectify everything.

→ The term “footpaths” is critical for study of the moons’ auroral footprints and we believe that it can not be replaced from the text without inducing confusion. Instead, we propose to define what a footpath is after the first use : “*The footpaths of the moons correspond to a sequence of magnetic projections of the moons’ location along their orbits onto Jupiter’s atmosphere, forming an oval-shaped contour. The moons’ footpaths therefore indicate the statistical location of the moons’ auroral footprints.* “

7. Line 282: “This strikingly compares to the source EDF for decametric emission by Io, Europa, and Ganymede.” Please elaborate on what the authors mean by the phrase “strikingly compares”.

→ We added a clarification at the end of the sentence : “*This is similar to the source EDF for decametric emission driven by Io, Europa, and Ganymede, i.e. an unstable loss-cone population of upgoing electrons*”

8. Line 324: Please add “... expanded magnetospheric state of the Jovian system.”

→ Done

9. Line 324: “This method could further be generalized to monitor...” What method are the authors referring to here. Please elaborate.

→ The discussion was to extend this method to Io, Europa, and Ganymede. We clarified the sentence accordingly : “*This method could further be generalized to Io, Europa, and Ganymede using observations of their TEB and MAW auroral spots to remotely monitor the properties of the current sheet, e.g. ion and electron density or plasma scale height.*”

10. Line 365: Please rewrite “...Callisto-magnetosphere interactions” to “...interactions of Callisto with Jupiter’s magnetosphere.”

→ Done

Reviewer #2

1. It would make sense to remind the reader why the violation of solid corotation of magnetospheric plasma occurs at a distance of ~ 20-40 RJ from the center of Jupiter (what is the reason) and why the footprints from the 3 Galilean satellites (Io, Europa and

Ganymede) are located equatorward from the main auroral oval, while the Callisto's one is poleward from it.

- **We agree with the reviewer that explaining why the corotation breakdown and main oval map to ~20-40 RJ is interesting. It is a balance of MIT coupling, strength of the internal magnetic field, effects of the current sheet field, and plasma properties. However, expanding on this topic is not mandatory in order to understand our analysis of the Callisto footprint. The central point is that the orbit of Callisto at R=26 RJ coincides with the mapping of the main oval. We therefore do not further describe the origin of corotation breakdown in the revised article.**
- **We have nonetheless added an explanation of the location of the moons' auroral footprints with respect to the main oval : *"Because the corotation breakdown is expected to occur at radial distances larger than the orbital distances of Io, Europa, and Ganymede, their UV and IR auroral footprints are located equatorward from the main auroral oval on Jupiter's polar regions."***

2 It is desirable to explain more clearly why the "increased outflow of plasma mass from the Io torus" is due to the expansion of the magnetosphere and why it leads to a shift of the main auroral oval equatorward? Now the reader is only referred to the specified references.

- **We have added a description of the processes leading to an equatorward shift of the main oval, introducing the stretching of the magnetic field line, as considered by theoretical models and observational studies.**

3. It is desirable to mention what kind of magnetospheric structure arises around Callisto when it is in the super-Alfvénic flow of Jupiter's magnetospheric plasma, which will make it impossible for this moon to generate auroras at the planet.

- **We precised in the text that in the case of a slightly-super-Alfvénic interaction, as experienced by Callisto when located at the center of the plasma sheet, no clear bow shock structure is formed (e.g. Ridley 2007, doi:10.5194/angeo-25-533-2007, simulated the formation of Earth's bow shock formation at $M_A \sim 8$). Still, even if a bow shock does not form, the energy transfer efficiency between the Callisto environment and Jupiter's polar region may be significantly reduced.**

4. It would be worthwhile to more clearly indicate what is the source of energy for generating the auroras from Callisto and the other Galilean satellites.

- **As it was not clearly stated in the text, we have added the following sentence in the introduction** *“While UV and IR auroral emissions are induced by electron precipitation within Jupiter’s atmosphere, moon-induced radio emissions result from unstable electron population propagating away from Jupiter.”*

Reviewer #3

Fig. 2a shows a background white dotted grid that is not explained. Presumably, it belongs to Jupiter’s body frame since it shows Callisto’s footpath cutting ~45 degrees across the high-latitude grid instead of being quasi-meridional. This could result from the tilted magnetic pole, which is the frame for the footpath. Does Juno’s relative motion, or the speed of the Alfvén wave (Extended Fig. 5) affect this footpath angle?

Adding an explanatory comment to the caption or text would be beneficial.

- **Thank you for this comment. Indeed, the white dotted grid is the Jupiter system III latitude/longitude grid, shown in a similar fashion as on Fig. 1b. We added a mention about it in the figure caption.**
- **The footpath angle you are mentioning is the magnetically mapped position of Callisto along its orbit. The position of Callisto is mapped following the combined JRM33 + KK2005 models, up to an altitude of 900 km above Jupiter 1-bar level (as stated in Methods’ “Magnetic Mapping” section). The latitude/longitude of the ultraviolet photons coming from the Callisto footprint, as recorded by Juno-UVS, were also computed assuming that they are coming from 900 km altitude above Jupiter’s 1-bar level. Since we are computing at 900 km altitude both (i) the predicted Callisto footpath position, and (ii) the latitude/longitude of the incoming Callisto footprint photons, we do not expect tomographic effects produced as Juno is travelling along its orbit.**
- **We added a mention regarding the altitude at which the UV photons are mapped (i.e., 900 km) in the Methods section “UV maps”.**

Extended Fig. 5: Is there a change in direction of the footpath where the propagation speed changes at the boundary of the high density equatorial region? Or is the footpath constrained at all propagation speeds by the tension in the magnetic flux line?

- **The change of density at the boundary of the high density equatorial region indeed affects the Alfvén speed, but does not affect the orientation of the satellite footpath. The Alfvén speed is defined as:**

$$v_a = \frac{B}{\sqrt{\mu_0 \rho}}$$

where ρ is the local plasma density, B the local magnetic field magnitude, and μ_0 the permeability of vacuum. Changing the density will only change the Alfvén travel time between Callisto and Jupiter. This will result in a delay in the arrival of the Alfvén waves generated at Callisto to reach Jupiter, compared to a case where the Alfvén waves travel at the speed of light. This difference between the “Magnetically mapped” position of the moon versus the “Alfvénically mapped” position of the moon (*i.e.*, travelling at the Alfvén speed) is the “Lead angle”, and is quite well-studied in the case of Io, Europa, and Ganymede (see, e.g., Hess et al. 2010, <https://doi.org/10.1016/j.pss.2010.04.011>; Hue et al. 2023, <https://doi.org/10.1029/2023JA031363>; Schlegel and Saur 2023, <https://doi.org/10.1029/2023JA031511>; Satoh et al. 2024, <https://doi.org/10.1029/2024GL110079>). However, because the Alfvén waves follow the magnetic field line, it will always arrive along the footpath calculated in this work, as shown as the solid and dashed green footpath contours shown in Fig. 2.

Line 260: “of in” Choose which word to use.

→ **Corrected**

Line 309: Need a comma after the closing parenthesis.

→ **Added**

Line 490: Define “CS” (current sheet?)

→ **We replaced this abbreviation by “current sheet”**

Line 533: f_{pe} is undefined (electron plasma frequency?)

→ **We added a definition f_{ce} and f_{pe} and their associated formula in the text**

Line 544: RX is undefined (Radio frequency?)

→ **RX stands for the right-handed extraordinary mode waves, this is now defined in the text**

Line 549: Where is f_h used?

→ **Thank you for spotting this mistake. F has been corrected by f_h in this equation.**

In situ and remote observations of the ultraviolet footprint of the moon Callisto by the Juno spacecraft

J. Rabia, V. Hue, C.K. Louis, N. André, Q. Nénon, J.R. Szalay, R. Prangé, L. Lamy, P. Zarka, B. Collet, F. Allegrini, R.W. Ebert, T.K. Greathouse, B. Bonfond, G.R. Gladstone, A.H. Sulaiman, W.S. Kurth, J.E.P. Connerney, P. Louarn, E. Penou, A. Kamran, D. Santos-Costa, R.S. Giles, J.A. Kammer, M.H. Versteeg, and S.J. Bolton

Response to the reviewer #2

We thank the reviewers for their careful review of our article. We respond hereafter to the latest reviewers' comments and we specify the changes made to the manuscript accordingly.

Does this mean that Callisto is outside the corotation breakdown, as its auroral footprint is located poleward of the main oval? If so, the corotation breakdown distance must be between 20 and 26 R_J in the case under consideration. If not, it would be desirable to clarify this issue.

→ **Yes, we clarified this point by completing the following sentence:** *“the expansion of the main emission results from a radial shift of the source of the emission, at radial distances below 26 R_J .”*

And what is the source of energy for this process?

→ **Acceleration of charged particles responsible for the multiwavelength auroral footprints is made possible by electric currents and/or Alfvén waves generated during the interaction between the plasma flow and the moons. We have added a sentence describing this process in the introduction :** *“Electrons responsible for these multiwavelength emissions are accelerated by Alfvén waves and/or electric currents generated by the local interaction between the magnetospheric plasma flow and the moons.”*